# LLM-Powered Preference Elicitation in Combinatorial Assignment

## Abstract

We study the potential of large language models (LLMs) as proxies for humans to simplify preference elicitation (PE) in combinatorial assignment, where the bundle space grows exponentially with the number of items, making full elicitation infeasible beyond small domains. Traditional elicitation methods sacrifice expressiveness and require agents to translate their preferences into rigid, unnatural formats, leading to under-reporting and welfare loss. Iterative, machine-learning-based elicitation schemes relax these constraints, but incur the cognitive burden of repeated, highly structured interaction. LLMs offer a one-shot alternative with reduced human effort. With the well-studied course-allocation problem as a model, we propose a framework for LLM proxies that can work in tandem with SOTA ML-powered preference elicitation schemes. We experimentally evaluate the efficiency of LLM proxies against human queries and investigate the model capabilities required for success. We find that our framework improves allocative efficiency by up to 20%, and these results are robust across different LLMs and to differences in quality and accuracy of reporting.

## 1 Introduction

*Preference elicitation (PE)* is central to effective decision-making. It must balance two objectives: (i) *minimizing* the cognitive burden on users by limiting how much they are asked to report, and (ii) *maximizing* the information extracted about their preferences for downstream decisions. PE arises in many domains; here we study it in the context of *mechanism design*.

In mechanism design, user preferences are high-dimensional and combinatorial, often suffering from the curse of dimensionality. Moreover, Nisan & Segal (2006) proved that for arbitrary value functions, achieving full efficiency in combinatorial auctions requires an exponential number of bids. To address this, most mechanisms in practice restrict users to report their preferences through structured languages. However, this limits the users' ability to fully articulate their preferences Nisan (2000); Sandholm & Suri (2000); Fujishima et al. (1999). Thus, the focus has shifted towards *iterative mechanisms*, where bidders interact with the mechanism over a series of rounds, providing only a limited amount of information in each interaction with the aim of maximizing the efficiency of the final allocation.

In this work, we focus on the *course allocation problem*, a well-studied combinatorial assignment problem (Budish, 2011) characterized by complex preferences (Budish & Kessler, 2021). In this problem, an educational institution must assign courses to students, who often have combinatorial preferences over course bundles, and there is limited seat availability in each course. Budish et al. (2017) introduced the *Course Match (CM)* mechanism, a significant improvement over previously existing approaches. CM has since been adopted by leading institutions such as Wharton and Columbia Business School University of Pennsylvania; Columbia Business School. However, CM's reporting language is both restrictive and cognitively demanding. As a result, students often make reporting errors, which negatively impact the mechanism's performance (Budish & Kessler, 2021). The iterative *MLCM* mechanism proposed by Soumalias et al. (2024a) addresses these limitations by allowing users to answer adaptively chosen *comparison queries (CQs)*. This alleviates reporting errors and leads to significant efficiency gains. While CQs are easy for students to answer, a long sequence of iterative queries can still pose a cognitive burden, and students remain inherently limited by the mechanism's reporting language.

We study the use of LLMs as proxies, answering queries for humans guided by a small amount of textual human input to reduce the reporting burden for the users and allow for a richer expression of preferences. For instance, consider the following illustrative description of preferences:

> "I prefer to take courses that are scheduled as closely together as possible so I can have an extra day off. If courses have a laboratory section, I strongly prefer that it be in the morning. Course A and Course B complement each other, and I would prefer to take them together to save time and effort..."

The textual description encodes combinatorial information about preferences without requiring commitment to any reporting language up-front, and it allows the students to easily express preferences over whole categories of courses (rather than labeling each one). It also provides a more natural elicitation process for students, who may find it easier to express their preferences in free text rather than answering tens of CQs. Our aim is to create a framework enabling a mechanism that requires structured input to also leverage such useful yet imprecise natural language input.

A proxy-based approach to preference elicitation has been explored previously, demonstrating the potential of LLMs to simulate human responses Horton (2023); Manning et al. (2024); Park et al. (2024). In concurrent work, Huang et al. (2025) explore the use of LLMs as proxies in combinatorial auction domains, adopting a different PE method that makes essential use of prices to coordinate elicitation (in contrast, our setting is one of assignment without money). For additional discussion of prior work on preference elicitation and machine learning in mechanism design, see Section A.

## 1.1 OUR CONTRIBUTIONS

In this work, we explore the potential of LLMs to improve and simplify preference elicitation in combinatorial assignment (specifically the problem of course allocation), and make three main contributions:

1. **One-shot elicitation mechanism.** We design an interface in which each user supplies a single narrative capturing scheduling constraints, content interests, and complementarities across courses. An LLM transforms this narrative into a latent value model that can be queried by an ML-powered preference elicitation framework.

2. **Robust proxy training.** To contend with variability in LLM outputs, we introduce (i) a *noise-robust loss* (Generalized Cross-Entropy Zhang & Sabuncu (2018)) within the mechanism that stabilizes the learning, (ii) acquisition functions (specifically Double Thompson Sampling (Dwaracherla et al., 2024)) that allow the mechanism to choose informative, low-variance queries during training.

3. **Empirical evaluation.** Through extensive experiments in course-allocation settings using a simulator fitted to real-world data (Soumalias et al., 2024a), we show through a simulated student model that this approach to elicitation achieves up to a 20% improvement in allocative value over CM. In comparison, the ML-based method of Soumalias et al. (2024a) achieves a 13.7% improvement using 20 manually answered comparison queries.

These results demonstrate that LLM-based one-shot proxies can outperform state-of-the-art iterative combinatorial assignment mechanisms, while offering a more natural and cognitively lightweight interface for users and remaining economically efficient for the platform. Looking forward, our framework can also be adopted to other ML-powered elicitation settings such as auctions (e.g., Brero et al. (2018); Soumalias et al. (2025)).

## 2 LLM-BASED PREFERENCE ELICITATION FRAMEWORK

Traditional preference elicitation methods face a trade-off: structured (one-shot) reporting languages must balance expressiveness and cognitive complexity, while iterative approaches, though more flexible, can lead to user fatigue. Our framework proposes a different paradigm: leveraging an LLM as a *one-shot proxy* for a user. This approach, depicted in Figure 1, aims to combine the expressiveness of natural language with the systematic learning capabilities of ML algorithms (which are used to adaptively query a latent value function implied by the natural language expression provided to the

LLM by a user), substantially reducing direct human effort. In this section, we describe the key components of our approach in more detail.

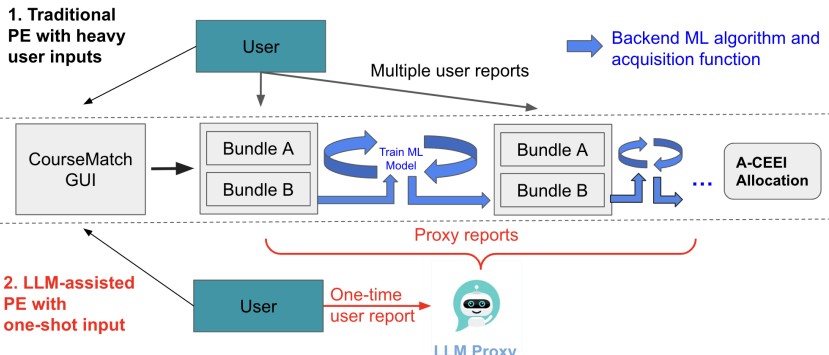

Figure 1: A schematic flowchart comparing Soumalias et al. (2024b) with the LLM proxy algorithm in this paper.

## 2.1 PROBLEM SETUP

We build on the MLCM mechanism (Soumalias et al., 2024a), which initializes a *cardinal* ML model of each student's value function based on her GUI reports. Following the Bradley-Terry model (Bradley & Terry, 1952), this cardinal model is further trained using the student's responses to pairwise *comparison queries (CQs)*. In MLCM, the CQs are determined by a specialized acquisition function that maintains an ordinal list of schedules based on the student's previous answers. Finally, an A-CEEI (Budish, 2011) is calculated, treating each student's ML model as her value function.

We extend MLCM by designing a framework that allows students to provide one-shot natural language input, enabling an LLM proxy to answer CQs on their behalf. This approach lets students express their preferences naturally and eliminates the need for iterative interaction with the mechanism. Additionally, the LLM proxy can handle a large number of CQs without imposing any cognitive burden on the student, unlocking potential efficiency gains. While our implementation focuses on course allocation, in Section 3.8 we discuss its applicability to other combinatorial allocation settings.

## 2.2 LARGE LANGUAGE MODELS AS USER PROXIES

### 2.2.1 NATURAL LANGUAGE PREFERENCE STATEMENT

At the heart of the framework is the user's *Preference Statement*: a single, free-form narrative that allows a user to express rich, multidimensional preferences, such as scheduling constraints (e.g., "no morning classes"), topical interests, and complementarities between courses, in a natural way.

Table 1 illustrates a six-course scenario. A user with such preferences might state:[1]

> *"Theory courses (Algorithms, ToC) top my list—ideally together. I avoid overlapping, applied topics like ML and NLP, and I'm not a morning person, so I prefer to avoid 9 AM lectures."*

This narrative succinctly conveys multiple preference dimensions that would be challenging to capture fully with structured GUIs or require multiple targeted comparision queries (CQs) to be answered.

### 2.2.2 DELEGATING COMPARISON QUERIES TO THE LLM PROXY

Given a user's preference statement, the LLM acts as a proxy on behalf of the user and no further interaction is required of the user. The ML-powered learning component poses to the LLM CQs of the form: "Is bundle $S_A$ preferred to bundle $S_B$, according to the student's textual preference

---

[1]Please see Appendix C.4 and Appendix C.5 for an example of the preferences of a valuation function for a student and examples of corresponding natural-language statements.

Table 1: Illustrative course attributes and pairwise interactions for a student. Positive interactions denote complements; negative interactions denote substitutes. The total value of a bundle $S$ for the student is the sum of base values for each course in $S$, plus pairwise interaction terms for any course pairs that co-occur in the bundle. "Time" denotes the scheduled lecture time for each course.

| Course | Time | Value | Interaction Matrix | | | | | |
|---|---|---|---|---|---|---|---|---|
| | | | Algo. | ToC | ML | NLP | DB | OS |
| Algorithms | 09:00 | 90 | 0 | 30 | 0 | 0 | 0 | 0 |
| Theory of Computation | 14:00 | 100 | 30 | 0 | 0 | 0 | 0 | 0 |
| Machine Learning | 15:30 | 50 | 0 | 0 | 0 | -30 | 0 | 0 |
| Natural Language Processing | 10:30 | 40 | 0 | 0 | -30 | 0 | 0 | 0 |
| Databases | 13:00 | 40 | 0 | 0 | 0 | 0 | 0 | -10 |

report?" The LLM, leveraging its interpretation of the student's preference statement and employing *chain-of-thought (CoT)* reasoning (Wei et al., 2022) to improve its inference, returns a response. This response is interpreted as an ordinal signal, e.g., $S_A \succ S_B$ or $S_B \succ S_A$.

### 2.3 ROBUST AND EFFICIENT PREFERENCE LEARNING

#### 2.3.1 MACHINE LEARNING MODEL

We follow recent papers on machine-learning powered combinatorial assignment (e.g., Weissteiner et al. (2023); Soumalias et al. (2024a;b; 2025)) in adopting a *monotone value neural network (MVNN)* (Weissteiner et al., 2022a) to approximate an agent's value function oon the basis of responses to CQs. MVNNs are a neural network architecture specifically designed for capturing value functions in combinatorial assignment problems, encoding prior information such as monotonicity of the value function at an architectural level. Before active elicitation, we initially train this network according to the CM GUI reports of the student.

#### 2.3.2 EFFICIENT QUERY SELECTION

This is an active learning setting, and to minimize LLM inference costs and accelerate learning we employ *Double Thompson Sampling (DTS)* as the acquisition function for selecting CQs. DTS has been used with great effectiveness in RLHF (Dwaracherla et al., 2024) and doubles our framework's efficacy compared to other acquisition functions (see Section 3.7). The use of DTS requires quantifying the learning algorithm's current uncertainty about an agent's true value function $v_i$. We achieve this by using an ensemble of MVNNs, termed an *epistemic MVNN (eMVNN)*, which represents a Bayesian belief distribution, $\mathcal{B}$, over possible value functions. At each round $t$ of the query process, DTS proceeds as follows:

1. Sample two candidate value functions for the agent, $v^{(1)}, v^{(2)} \sim \mathcal{B}$, from the eMVNN.

2. Identify the predicted optimal feasible bundles for the agent with respect to the two candidate value functions, $S_t^{(1)} \in \arg\max_{x \in \mathcal{F}} \hat{v}^{(1)}(x)$, $S_t^{(2)} \in \arg\max_{x \in \mathcal{F}} \hat{v}^{(2)}(x)$.

3. Repeat Steps (1) and (2) until $S_t^{(1)} \neq S_t^{(2)}$.

The query $(S_t^{(1)}, S_t^{(1)})$ is then posed to the LLM proxy. This acquisition function quickly focuses on the most important part of the allocation space: bundles that are valuable yet attainable for the agent, and where there is disagreement according to the Bayesian belief as to which bundle is optimizing for the agent.

#### 2.3.3 ROBUST LEARNING FROM NOISY LLM RESPONSES

To combine the cardinal information provided by a student's GUI reports with the ordinal information provided by the LLM proxy's responses $\tilde{r}_t$ to comparison queries $(S_t^{(1)}, S_t^{(2)})$, we adopt the mixed training approach of Soumalias et al. (2024a), applying a regression loss to the former and a classification loss to the latter. Unlike in prior work, though, the LLM proxy's responses are inherently noisy approximations of the student's true preferences.

To train the MVNN $\hat{v}_i$ robustly in the presence of this noise, we minimize the *Generalized Cross-Entropy (GCE)* loss (Zhang & Sabuncu, 2018), defined as $\mathcal{L}_\gamma(\mathbf{p}, \mathbf{y}) = \frac{1 - \mathbf{y} \cdot \mathbf{p}^\gamma}{\gamma}$, where $\mathbf{y}$ is the one-hot encoded label inferred from the LLM proxy's response (e.g., if the LLM prefers $S_k$ over $S'_k$, then $\mathbf{y} = (1, 0)$), and $\mathbf{p}$ is the model-predicted distribution over preferences (e.g., $p(S_k \succ S'_k) = \sigma(\hat{v}(S_k) - \hat{v}(S'_k))$), with $\sigma$ the sigmoid function. The hyperparameter $\gamma \in (0, 1]$ controls robustness to label noise: GCE recovers cross-entropy as $\gamma \to 0$ and mean absolute error as $\gamma \to 1$. With this loss, we can prove:

**Proposition 2.1.** *If the LLM proxy's accuracy in answering CQs (relative to the user's true value function $v_i$) is strictly greater than $50\%$, then the user's true value function $v_i$ is a minimizer of the expected GCE loss calculated over the noisy dataset of CQs and LLM responses when learning $\hat{v}_i$.*

The proof is deferred to Appendix B. As long as the LLM proxy is a better-than-random predictor of the user's preferences, the GCE loss guides the learning of $\hat{v}_i$ towards the true value $v_i$.

As reported in Section 3, the LLM proxies used in our experiments achieve accuracy above this threshold (e.g., $\geq 71\%$), making this theoretical observation relevant to our practical results. [2]

## 3 EXPERIMENTS

### 3.1 EXPERIMENT SETUP

We use the Course Allocation Simulator from Soumalias et al. (2024a), as it offers a unique combination of extensive preference data and validated error rates from real lab experiments. Specifically, its configuration closely mirrors the results in Budish & Kessler (2021), matching both the frequency and severity of student mistakes within a 1% margin. To focus solely on learning performance, we set all course capacities to infinity, ensuring that each student can receive any combination of five courses. In line with our LLM-based design, we first convert each synthetic student's cardinal preferences into a free-text description using Llama-3.1. These narratives are then passed to the LLM proxy, which answers comparison queries by interpreting the student's stated preferences. Further implementation details, including our hyperparameter optimization (HPO) protocol, are provided in Section D.2.

### 3.2 EFFICIENCY RESULTS

In Figure 2a we plot the student's allocated bundle value against the number of CQs answered by her proxy LLM.[3] We normalize the value of the bundle each student receives by the bundle that the student would have received based on her initial GUI reports. Our results demonstrate that, with just one natural language input from the student, our framework increases allocative value by 19.3%. Furthermore, the student prefers her allocation under our framework in 74% percent of the cases. For context, in this same simulation, MLCM, the SOTA mechanism for course allocation, increases allocative efficiency by 14.2% using 20 student-answered CQs. This highlights the effectiveness of our framework in improving allocation outcomes with significantly lower student effort.

### 3.3 LEARNING PERFORMANCE

To assess our framework's impact on the generalization performance of a student's ML model, in Figure 2b we plot ($\text{MAE}_C$), a shift invariant measure of the mean absolute error against the number of LLM-answered CQs.

We focus on a shift-invariant regression metric because the allocation algorithm assigns to each student the bundle with the highest predicted value–based on her ML model–that is attainable for her. Thus, learning a student's value function up to a constant shift suffices, since it results in the same (optimal) allocation as the true value function under any set of constraints. Similarly, we focus on the student's top quantile of bundles, as these are the most critical for ensuring high-value allocations.

---

[2] In Section 3.6 we experimentally show that employing GCE instead of the standard BCE used in Soumalias et al. (2024a) doubles the efficacy of our framework.

[3] Note that in our framework, each student provides only a single piece of natural language input in a one-shot manner. Thus, the x-axis reflects the computational cost for the mechanism rather than the cognitive effort required from the student.

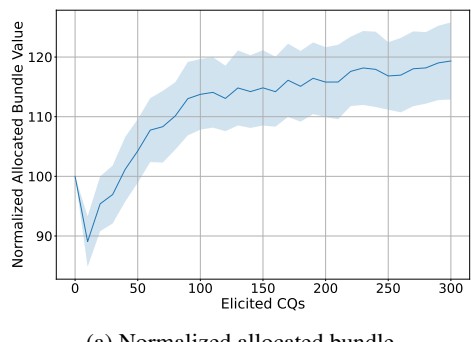

(a) Normalized allocated bundle

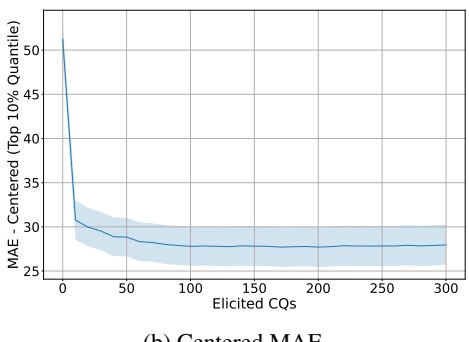

(b) Centered MAE.

Figure 2: Comparison of (a) normalized allocated bundle value and (b) centered MAE, both as functions of the number of LLM-answered CQs. Shown are averages over 100 instances including 95% CIs.

| Metric | Whole Test Set | | Top 10% Quantile | |
|---|---|---|---|---|
| | **Before** | **After** | **Before** | **After** |
| $\mathrm{MAE}_C$ | 46.14 | 30.69 | 51.23 | 27.96 |
| KT | 0.266 | 0.389 | 0.121 | 0.283 |
| $\mathrm{MSE}_C$ | 3475 | 1942 | 4172 | 1500 |
| $R^2_C$ | $-0.62$ | 0.22 | $-2.50$ | 0.14 |

Table 2: Comparison of learning metrics before and after applying our framework. Each metric is reported for the whole test set and the top 10% quantile in terms of student value. Shown are averages over 100 runs. Winners based on a paired t-test with $\alpha = 1\%$ marked in gray.

Our results demonstrate that our framework leads to an immediate and significant reduction in $\mathrm{MAE}_C$, nearly halving its original value.

In Table 2, we further analyze the impact of our framework on the generalization performance of a student's ML model using both rank-based and shift-invariant learning metrics. We report results for the top quantile to assess performance in the most consequential part of the bundle space, as well as across the entire bundle space to measure overall generalization. In Section D.3 we present an expanded version of this table and find similarly strong results for more quantiles.

Table 2 demonstrates that our framework—using just a *single* natural language input from the student—substantially enhances all learning metrics, both rank-based and shift-invariant. This improvement is even more pronounced in the top quantile, the most critical region from a Bayesian optimization perspective. Moreover, the performance gains achieved by our framework are statistically significant at the $\alpha = 1\%$ confidence level for *all* metrics and quantiles.

Taken together, our results in Sections 3.2 and 3.3 show that our LLM-powered PE framework leads to vastly improved learning performance and allocative efficiency.

### 3.4 ROBUSTNESS TO STUDENTS' MISTAKES

In this section, we evaluate our framework's performance when changing the severity of the students' mistakes in the GUI language. To do this, we multiply all parameters affecting their mistake profile in the simulator by a constant $\gamma$. For $\gamma > 1$, students make more mistakes than in the default profile, and for $\gamma < 1$ the opposite is true. Importantly, we do not change any of the hyperparameters of our framework compared to their optimized values for our default setting.

In Table 3, we present the normalized allocation value, and percentage of instances where our framework increases allocative value (over just the GUI reports) as a function of the mistake multiplier. We observe that our framework improves average allocation even for $\gamma = 0.5$. Importantly, mistakes do not scale linearly; in that case, students make about half the amount of mistakes reported in

Budish & Kessler (2021), and the severity of those mistakes is reduced by over 80% (Soumalias et al., 2024a). Overall, these results demonstrate that our framework is robust to varying mistake severity, consistently improving allocation outcomes even under significantly reduced or amplified error profiles.

| Noise Mult. | Normalized Value $\pm CI$ | % of Runs | | P-value |
|---|---|---|---|---|
| | | Better | Worse | |
| 0.5 | $103.93 \pm 6.76$ | 52 | 48 | 0.132 |
| 0.75 | $108.66 \pm 7.31$ | 66 | 34 | 0.013 |
| 0.9 | $106.08 \pm 8.22$ | 54 | 46 | 0.079 |
| 1 | $119.34 \pm 6.46$ | 74 | 26 | 3.41e-8 |
| 1.1 | $113.05 \pm 8.39$ | 66 | 34 | 0.002 |
| 1.25 | $122.64 \pm 8.12$ | 74 | 26 | 9.37e-7 |

Table 3: Normalized allocated bundle value with our framework as a function of the noise multiplier used. Shows are averages over 100 runs over 100 runs for the default noise level and 50 runs for all others, including 95% CIs. We also show the % of cases where our framework improves a student's allocation, and the significance of these results.

### 3.5 LLM ARCHITECTURE ABLATION TEST

In this section, we evaluate our framework's robustness to LLM architectural variations. As detailed in Section 2, our experiments employ two distinct LLMs: the LLM proxy used by our framework to finetune the student's ML model, and the LLM simulating student textual input to our framework.

Table 4 presents performance results across different LLM architectures for both student simulation and CQ answering. The results demonstrate remarkable consistency: allocation value improves by 19–22% (with high statistical significance) across all configurations, with student allocations improving in 80–82% of cases. This robust performance stems from consistently high CQ accuracy (72–75%) by the LLM proxy across all configurations. As expected, commercial, larger models outperform open-sourced smaller models, though we primarily use the latter for cost efficiency. These findings demonstrate our framework's adaptability to both architectural variations and diverse natural language inputs, with further analysis of CQ accuracy sensitivity presented in Section 3.8.

| LLM Surrogate | LLM Proxy | Allocation Value $\pm CI$ | % Better | CQ Accuracy (%) | P-value |
|---|---|---|---|---|---|
| ChatGPT | ChatGPT | $120.67 \pm 8.44$ | 82 | 72.11 | 9.84e-6 |
| LLaMA | ChatGPT | $121.72 \pm 8.76$ | 82 | 73.70 | 8.38e-6 |
| LLaMA | LLaMA | $119.34 \pm 6.50$ | 74 | 71.69 | 6.8e-8 |

Table 4: LLM-powered PE framework performance under different LLM architectures. Each setup involves one LLM acting as a surrogate student, simulating the student's natural language input to the mechanism and another LLM proxy answering CQs based on the input. We report the normalized allocated bundle value, the percentage of instances where our framework improves allocative efficiency, the CQ accuracy of the CQ answerer LLM, and the statistical significance of the allocated bundle value results (p-value). Shown are averages over at least 50 runs for each configuration, including 95% confidence intervals (CIs).

### 3.6 COMPARING BCE AGAINST GCE

In this section, we study the effect of the GCE loss in improving our framework's performance. We compare allocation value under the default model parametrization (Section D.2) with a version using standard binary cross-entropy (BCE) loss. Both versions were allocated the same compute time for hyperparameter optimization (HPO), as detailed in Section D.2.

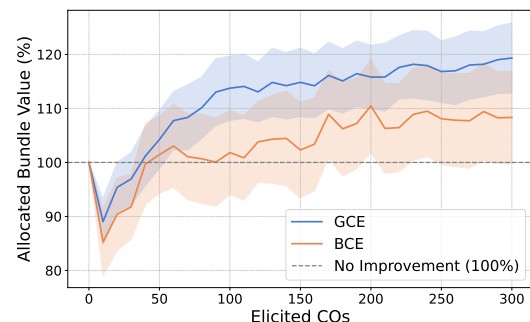

Figure 3: Normalized allocated bundle value as a function of the number of LLM-answered CQs. We compare GCE versus BCE as the training loss for the LLM-answered CQs. Shown are averages over 50 instances including 95% CIs.

Similar to our main results in Section 3.2, in Figure 2a we plot the student's allocated bundle value, as a function of the number of CQs answered by the student's LLM proxy. We observe that for any number of CQs, GCE outperforms BCE. Importantly, GCE increases allocation value by an average of 19.3% using 300 CQs, while BCE only achieves an increase of 8.3%, and GCE has a much more significant p-value ($6.84 \times 10^{-8}$).

### 3.7 COMPARING DIFFERENT ACQUISITION FUNCTIONS

In this section, we evaluate the importance of the acquisition function in our framework's performance. We compare the allocation value across different acquisition functions, using the default parametrization of our framework as described in Section D.2, while varying only the acquisition function responsible for selecting the CQs.

Table 5 highlights the effectiveness of Double Thompson Sampling, which doubles the improvement in allocation value compared to the second-best acquisition function—from 10.95% to 19.34%. Notably, Double Thompson Sampling is also recognized as a superior acquisition function in RLHF settings (Dwaracherla et al., 2024). Interestingly, all other acquisition functions tested, including random selection, exhibit nearly identical performance.

| Acqu. Function | Normalized Value $\pm CI$ | % Runs | | P-value |
| --- | --- | --- | --- | --- |
| | | **Better** | **Worse** | |
| DoubleTS | $119.34 \pm 6.50$ | 74 | 26 | 6.8e-8 |
| Infomax | $110.51 \pm 8.05$ | 66 | 34 | 0.014 |
| Boltzmann | $110.95 \pm 8.86$ | 60 | 40 | 0.019 |
| Random | $109.86 \pm 9.00$ | 64 | 36 | 0.037 |

Table 5: Normalized allocated bundle value with our framework for various acquisition functions. Shows are averages including 95% CIs. We also show the % of cases where our framework improves a student's allocation, and the significance of these results.

### 3.8 LLM ACCURACY ROBUSTNESS TEST

In this section, we evaluate our framework's robustness to variations in the accuracy of the proxy LLM. Given the consistently high performance (72–75%) of tested LLM architectures (Section 3.5), we employ *simulated LLMs* to explore a broader accuracy range. Specifically, for each CQ chosen by the acquisition function, we provide the correct reply with probability equal to the simulated LLM accuracy, assuming i.i.d. mistakes.[4] Importantly, for this test, we do not perform HPO, but instead use the values for all hyperparameters determined in our default setting.

---

[4]Figure 4 motivates this choice, as the LLM accuracy remains roughly constant throughout the process.

Table 6 demonstrates our framework's robustness: it continues to improve allocation value even when LLM accuracy drops to 60%. This robustness is particularly noteworthy given that all tested LLMs maintain 72–75% accuracy (Section 3.5). For comparison, the state-of-the-art mechanism by Soumalias et al. (2024a) loses half its efficacy with just 16% student error rate in CQ answers.[5]

| LLM CQ Accuracy | Normalized Value $\pm CI$ | % of Runs | | P-value |
|---|---|---|---|---|
| | | Better | Worse | |
| 70% | $126.50 \pm 1.79$ | 84.5 | 14.6 | 2.6e-135 |
| 65% | $118.03 \pm 1.85$ | 71.9 | 27.6 | 7.5e-70 |
| 60% | $104.29 \pm 1.75$ | 52.9 | 47.1 | 8.93e-7 |
| 55% | $85.76 \pm 1.65$ | 26.1 | 73.9 | 1.0 |

Table 6: Normalized allocated bundle value with our framework for various levels of *simulated* LLM accuracy. Shows are averages over 1000 runs including 95% CIs. We also show the % of cases where our framework improves a student's allocation, and the significance of these results.

In Appendix, we provide all prompts we used, example outputs, and an additional discussion of the effects of brevity and quality of student input on our framework's performance (Sec. C.5.2).

In our experiments, each student required roughly 300 comparison queries, amounting to an average of 0.1 million input tokens and nearly the same amount of output tokens. Even using the commercial models like GPT-4o, this is roughly only 1 dollar in cost. Our experiments further show that we are not reliant on advanced models, and even using open-sourced models with smaller parameter sizes like LLaMA 3.1 8b, we are still able to achieve significant results. With both the capabilities and cost structures of LLMs *rapidly evolving*, we imagine cheaper reasoning-based models achieving better results at around the same cost.

Although our experiments focus on the course allocation domain, the same LLM-based one-shot preference elicitation approach naturally extends to a wide array of settings in which agents have complex preferences over combinatorial outcomes. Our framework is broadly applicable to other iterative combinatorial allocation mechanisms that use trained ML models to guide query generation, such as MLCA, ML-CCA, and BOCA (Brero et al., 2021; Soumalias et al., 2024a; Weissteiner et al., 2023); as detailed in Section 3.3, our framework improves the generalization of agents' ML models, particularly in critical regions shown to enhance allocative outcomes (Soumalias et al., 2025). As LLMs continue to improve in both capability and affordability, they present a promising new approach for making mechanisms more practical and accessible in real-world applications.

## 4 CONCLUSION

We addressed the fundamental challenge in combinatorial preference elicitation: the trade-off between expressive power and the high cognitive burden placed on users. Our work demonstrates that large language models can serve as effective one-shot proxies, translating a single, natural language narrative from a user into the structured inputs required by complex assignment mechanisms. We introduced a robust framework that handles the inherent noise of LLM responses using a GCE loss and efficiently learns preferences through an active learning strategy powered by Double Thompson Sampling. Our experiments in the course allocation domain, using a simulator fitted to real-world data, show that this approach improves allocative efficiency by up to 20% over strong baselines, with results holding across various LLM architectures and input qualities.

This work opens avenues for designing more user-centric mechanisms in other complex domains like combinatorial auctions and resource allocation. Future research could explore integrating more sophisticated LLM reasoning or conducting large-scale human-subject studies to further validate this promising paradigm for preference elicitation.

---

[5]In their mechanism, the CQs are answered by real students, and prior work suggests that near-perfect accuracy by real students can generally be assumed.

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

APPENDIX

## A   PRIOR WORK

**Preference elicitation and learning theory**   Blum et al. (2004) and Lahaie & Parkes (2004) first frame the preference elicitation problem in terms of learning theory, adapting tools for learning

boolean functions from few queries to the problem of learning valuation functions. Lahaie & Parkes (2004) emphasize the goal of determining the efficient allocation, and not just accurate elicitation for the sake of elicitation. Many other works build in the same direction (Conitzer, 2007; Balcan et al., 2012; Zhang & Conitzer, 2020; Lock et al., 2022).

**Iterative auctions with machine learning**   Some auctions ask for bids in multiple rounds, such as the *combinatorial clock auction (CCA)* (Ausubel et al., 2006), which generated over *USD* 20 *billion* in revenue from high-stakes spectrum auctions between 2012 and 2014 (Ausubel & Baranov, 2017). More sophisticated iterative mechanisms attempt to make optimal queries to maximize allocative efficiency while minimizing the reporting burden on users. Brero et al. (2017; 2018) introduced this technique, using support vector regression to model bidders' value functions. Brero et al. (2021) generalized these techniques and applied them to a realistic combinatorial domain motivated by spectrum auctions, and showed better allocative efficiency than the CCA.

Other authors have improved on this work by using neural networks (Weissteiner & Seuken, 2020), by learning value functions in the Fourier domain (Weissteiner et al., 2022b), by imposing monotonicity constraints on the neural network value functions (Weissteiner et al., 2022a), and by using estimations of uncertainty to choose queries in an active learning style (Weissteiner et al., 2023). More recently, this framework has been adapted to use demand queries instead of value queries, similar to the interaction paradigm of the CCA (Soumalias et al., 2024b). Soumalias et al. (2025) showed how both query types can be leveraged to create the most query-efficient ICA to date.

**Machine learning for mechanism design**   Machine learning, particularly deep learning, has been increasingly applied to mechanism design. Dütting et al. (2023) introduced deep neural networks for auction design, improving representational flexibility. This approach, termed *differentiable economics*, has since been extended to budget-constrained bidders (Feng et al., 2018), payment minimization (Tacchetti et al., 2019), multi-facility location (Golowich et al., 2018), fairness-revenue trade-offs (Kuo et al., 2020), two-sided matching (Ravindranath et al., 2021), and data markets (Ravindranath et al., 2023). Structural advancements, such as Wang et al. (2024), achieve exact strategy-proofness rather than approximate incentive compatibility. It is also recently extended to the single-bidder combinatorial auction setting (Wang et al. (2025)). Unlike the present work, this line of work focuses on searching through the space of mechanisms, rather than using ML for preference elicitation in a fixed mechanism. Moreover, with the exception of Wang et al. (2025) these methods address settings with additive and unit-demand valuations, leaving problems with combinatorial valuations, such as the course allocation problem, largely unaddressed.

**LLMs for Translating Natural-Language Descriptions to Formal Game Models**   A related thread of research explores how LLMs can interpret free-form textual narratives and translate them into formal, game-theoretic structures (Daskalakis et al., 2024; Mensfelt et al., 2024a;b; Deng et al., 2025). Mensfelt et al. (2024a;b) map textual inputs to logic-based representations of smaller or simultaneous-move games, Daskalakis et al. (2024) address more complex, story-like sequences by constructing extensive-form games that allow subsequent equilibrium analysis, and Deng et al. (2025) translate game descriptions in natural language into game-theoretic extensive-form representations. While these works focus on different problem settings than course allocation, they illustrate broader opportunities for harnessing LLMs to extract actionable, structured information from informal descriptions—an approach that aligns with our use of free-text inputs to inform combinatorial assignments.

## B    DETAILS FROM SECTION 2

### B.1   OMMITED PROOFS

*Theorem 2.1 Proof.*   Ghosh et al. (2015) showed that, under uniform label noise with a noise rate $\eta \leq \frac{c-1}{c}$, where $c$ is the number of classes, the minimizer of a symmetric loss function over a noiseless dataset is also the minimizer of the same loss function over a noisy dataset. Since $c = 2$ in our case, label noise is inherently uniform and this requirement becomes $\eta \leq 0.5$.

The GCE loss is known to be symmetric Zhang & Sabuncu (2018), so the student's true valuation function minimizes the GCE loss on the noisy dataset generated by the LLM responses.    $\square$

**Input**: $\{X_{reg}, y_{reg}\}, \{X_{class}, y_{class}\}$
**Parameters**: epochs $t_{reg}$, learning rate $\eta_{reg}$, regularization parameter $\lambda_{reg}$, epochs $t_{class}$, learning rate $\eta_{class}$, regularization parameter $\lambda_{class}$
**Output**: Parameters of trained ML model $\mathcal{M}$

1: $\theta_0 \leftarrow$ initialize parameters of the ML model $\mathcal{M}$
2: **for** $i = 1$ to $t_{reg}$ **do**
3:    $loss_{reg} \leftarrow 0$
4:    **for** each $(x, y)$ in $\{X_{reg}, y_{reg}\}$ **do**
5:       $\hat{y} \leftarrow \mathcal{M}^{\theta_{i-1}}(x)$
6:       $loss_{reg} \leftarrow loss_{reg} + l_{reg}(y, \hat{y}) + \lambda_{reg} \sum \theta_{i-1}^2$
7:    **end for**
8:    $\theta_i \leftarrow \text{ADAM}(\theta_{i-1}, loss_{reg}, \eta_{reg})$
9: **end for**
10: **for** $i = t_{reg} + 1$ to $t_{reg} + t_{class}$ **do**
11:    $loss_{class} \leftarrow 0$
12:    **for** each $((x_1, x_2), y)$ in $\{X_{class}, y_{class}\}$ **do**
13:       $\hat{y}_1 \leftarrow \mathcal{M}^{\theta_{i-1}}(x_1)$
14:       $\hat{y}_2 \leftarrow \mathcal{M}^{\theta_{i-1}}(x_2)$
15:       $\hat{y} \leftarrow \frac{1}{1 + e^{-(\hat{y}_1 - \hat{y}_2)}}$
16:       $loss_{class} \leftarrow loss_{class} + l_{class}(y, \hat{y}) + \lambda_{class} \sum \theta_{i-1}^2$
17:    **end for**
18:    $\theta_i \leftarrow \text{ADAM}(\theta_{i-1}, loss_{class}, \eta_{class})$
19: **end for**
20: **return** $\theta_{t_{reg} + t_{class}}$

**Algorithm 1:** Mixed training for regression model $\mathcal{M}$

### B.2   Mixed Training Algorithm and Connection to the Bradley-Terry Model

In this section, we reprint the training algorithm of Soumalias et al. (2024a) for integrating GUI reports (regression data) and CQs (ordinal data) into the training of MVNNs.

The core idea of Algorithm 1 is to first train the ML model on the student's GUI reports, and then finetune that training on her CQ responses. During the regression phase, the algorithm uses a typical regression loss, which compares the real-valued output of the model, $\mathcal{M}(\cdot)$, to the inferred value from the student's GUI input for a particular schedule (Line 6).

In contrast, for CQs, we transform the model's real-valued outputs for two schedules, $x_1$ and $x_2$, using the sigmoid function $f(x) = \frac{1}{1 + e^{-x}}$. This yields a predicted probability within the $[0, 1]$ range, where $\frac{1}{1 + e^{-(\hat{y}_1 - \hat{y}_2)}}$ represents the likelihood that the student finds schedule $x_1$ preferable to schedule $x_2$ (Line 15). Note that this predicted probability is exactly the one under the Bradley-Terry model (Bradley & Terry, 1952). This predicted probability is then compared with the actual binary preference expressed by the student: 1 if she preferred $x_1$, and 0 otherwise (Line 16).

The training process runs for a total of $t_{reg} + t_{class}$ epochs, after which the final set of parameters is returned (Line 20).

### B.3   Monotone Value Neural Networks (MVNNs)

In this section, we reporting the original definition of MVNNs introduced in Weissteiner et al. (2022a):

**Definition B.1** (MVNN). An MVNN $\mathcal{M}_i : \mathcal{X} \to \mathbb{R}_{\geq 0}$ for agent $i \in N$ is defined as

$$\mathcal{M}_i(x) \coloneqq W^{i,K_i} \varphi_{0, t^{i, K_i - 1}} \left( \ldots \varphi_{0, t^{i,1}} (W^{i,1} (Dx) + b^{i,1}) \ldots \right) \tag{1}$$

- $K_i + 2 \in \mathbb{N}$ is the number of layers ($K_i$ hidden layers),

- $\{\varphi_{0, t^{i,k}}\}_{k=1}^{K_i - 1}$ are the MVNN-specific activation functions with cutoff $t^{i,k} > 0$, called *bounded ReLU (bReLU)*:

$$\varphi_{0, t^{i,k}}(\cdot) \coloneqq \min(t^{i,k}, \max(0, \cdot)) \tag{2}$$

- $W^i := (W^{i,k})_{k=1}^{K_i}$ with $W^{i,k} \geq 0$ and $b^i := (b^{i,k})_{k=1}^{K_i-1}$ with $b^{i,k} \leq 0$ are the *non-negative* weights and *non-positive* biases of dimensions $d^{i,k} \times d^{i,k-1}$ and $d^{i,k}$, whose parameters are stored in $\theta = (W^i, b^i)$.

- $D := \text{diag}\left(\frac{1}{c_1}, \ldots, \frac{1}{c_m}\right)$ is the linear normalization layer that ensures $Dx \in [0, 1]$ and is not trainable.

*Remark* B.2 (Initiaization). We always use the initialization scheme from Weissteiner et al. (2023, Section 3.2 and Appendix E), which offers crucial advantages over standard initialization schemes as discussed in Weissteiner et al. (2023, Section 3.2 and Appednix E).

### B.4 EPISTEMIC MVNNS (EMVNNS)

In this section, we introduce *Epistemic MVNNs (eMVNNs)*, which extend the standard MVNN architecture by incorporating an ensemble-based method to estimate epistemic uncertainty. Epistemic uncertainty reflects the model's lack of knowledge about certain regions of the input space and is crucial for guiding active learning and preference elicitation tasks. We formally define Epistemic MVNNs below.

**Definition B.3** (Epistemic MVNNs). An *Epistemic MVNN* for agent $i \in N$ is defined as an ensemble of $M$ independently initialized MVNNs, denoted by $\{\mathcal{M}_i^{(j)} : \mathcal{X} \to \mathbb{R}_{\geq 0}\}_{j=1}^M$, where each network is defined as in Theorem B.1. For a given bundle $x \in \mathcal{X}$, the prediction distribution is represented by the set of outputs from the ensemble:

$$\mathcal{M}_i^E(x) = \{\mathcal{M}_i^{(j)}(x) : j = 1, \ldots, M\}.$$

The *mean prediction* and *epistemic uncertainty* are defined as follows:

1. **Mean prediction:**

$$\hat{y}_i(x) = \frac{1}{M} \sum_{j=1}^M \mathcal{M}_i^{(j)}(x),$$

2. **Epistemic uncertainty:**

$$\sigma_i^2(x) = \frac{1}{M} \sum_{j=1}^M \left(\mathcal{M}_i^{(j)}(x) - \hat{y}_i(x)\right)^2.$$

Here, $\sigma_i(x)$ denotes the standard deviation of the ensemble's predictions and serves as a measure of epistemic uncertainty.

*Remark* B.4. Motivated by the connection of our problem to RLHF highlighted in Section 2, we use $M = 10$ following (Dwaracherla et al., 2024).

*Remark* B.5. Note that the mean prediction of an eMVNN preserves the structural properties of MVNNs. Specifically:

- **Zero value for the empty bundle:** Since each MVNN in the ensemble is designed to map the empty bundle to zero, the mean prediction, as a convex combination of these outputs, also assigns a zero value to the empty bundle.

- **Monotonicity:** Each MVNN is a monotone function by construction, as its weights are constrained to be non-negative. The mean of monotone functions is itself monotone, ensuring that the mean prediction of the eMVNN is also monotone.

# C DETAILS OF LLM PROXIES

prompt here, amount of CoT, etc.

## C.1 ACCURACY OF LLM CQS OVER ROUNDS

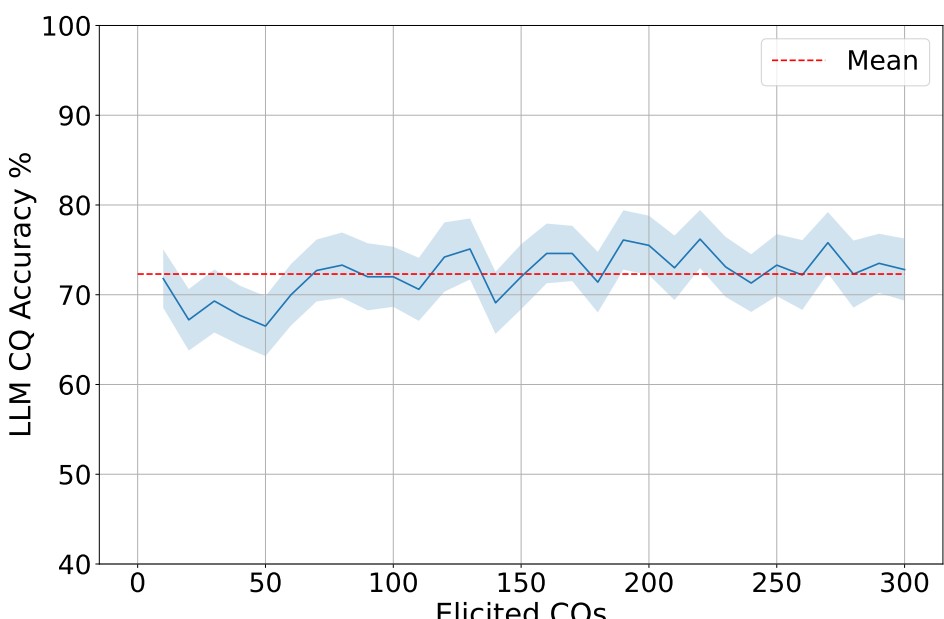

Figure 4: LLM CQ accuracy (evaluated using LLaMA 3.1 as the language model) as a function of the number CQs already answered by the LLM. Note that later CQs are expected to be both more informative and harder to answer, as they are generated based on an acquisition function that takes into account both the current ML model, and its epistemic uncertainty. Nonetheless, accuracy remains high even with many queries. Shown are averages over 100 instances including 95% CIs.

In Figure 4, we observe that the LLM proxy accuracy remains relatively constant throught the ML model's finetuning on CQs. Note that CQs are not chosen independently; instead, queries are chosen based on acquisition functions (Dwaracherla et al., 2024) that take into account both the student's ML model and its epistemic uncertainty. Thus, later queries are expected to be more informative and harder to answer since the student's value model has been already improved based on the previous queries.

## C.2 STUDENT PREFERENCE CONVERSION PROMPT

```
Please act as a student describing their course preferences for the
    upcoming semester. Write a detailed, first-person paragraph about
    your preferences based on the following information:

Course Preferences (ordered from highest to lowest value within each
    tier):

High Priority Courses:
{format_tier_courses(high_pref)}

Medium Priority Courses:
{format_tier_courses(medium_pref)}

Lower Priority Courses:
{format_tier_courses(low_pref)}

Course Relationships:
```

```
Overlapping Content (Substitutes) – These are courses that HARMS YOU
    when taken together compared to taking only one, and the more the
    worse:
{chr(10).join(f"- {group}" for group in substitute_groups)}

Complementary Courses (Complements) – These are courses that BENEFITS
    YOU HARMS YOU when taken together compared to taking only one, and
    the more the better:
{chr(10).join(f"- {group}" for group in complement_groups)}

Additional Constraints:
- Budget Constraint: {student.budget:.2f}
{f"- Time Gap Penalty: {student.timegap_penalty}" if
    student.timegap_penalty != 0 else ""}
{f"- Overload Penalty: {student.overload_penalty}" if
    student.overload_penalty != 0 else ""}

Please write a natural, detailed explanation of these preferences as if
    you were the student. Include:
1. Your strongest course interests, explaining them in order of
    preference within each priority tier
2. How you're thinking about course combinations, discussing specific
    synergies and overlaps:
    - When describing overlapping courses, explain how much the overlap
    affects your interest
    - When describing complementary courses, explain how much additional
    value you see in taking them together
3. Your overall strategy for course selection, considering both your
    budget constraints and the strength of course relationships
4. Any specific scheduling or workload considerations

Keep the tone conversational and authentic to how a student would
    describe their course preferences. Make sure to reference both your
    relative preferences within each tier and the specific impacts of
    course combinations on your overall academic plan. Your response
    should be three paragraph: the first paragraph should list all the
    top and medium tier courses you want to pick, and any courses that
    comes with each that might be complementary (good) or subtitutes
    (bad). The second paragraph should clearly and detailedly state ALL
    OF THE bundles of courses that are complements and bundles of
    courses that are substitutes for you; for each, explain how much it
    hurts you when you take different numbers of courses from that
    bundle. You should cover all the complement bundles and substitute
    bundles given to you – leave nothing out. The thrid paragraph
    concludes and highlights other concerns. Aim for qualitatively
    detailed description and avoid saying exact numerical values in the
    output (e.g. it's fine to say 'taking x and x together is
    suboptimal, taking x, x and x together even more suboptimal, and
    taking x,x,x, and x together should really be avoided, (so on so
    forth until the size of the complement/substitute set).', but not
    fine to say 'My value for course 16 is 102.45', or 'x and x together
    decreases my utility by 43%'.)
"""
```

## C.3 COMPARISON QUERY PROMPT

```
Based on these student preferences:

{student_preferences_text}

Compare:
Bundle A: {format_bundle(bundle1)}
Bundle B: {format_bundle(bundle2)}
```

```
to choose the better bundle.

Please ignore budget constraint if it's mentioned in the preference -
    pretend it doesn't exist.

Your response must use these EXACT tags below, and ONLY include the
    tags, end your response after that. The text between tags should be
    concise.

'''
<PREFERENCES>
Bundle A: [First recall the courses in Bundle, then list matching
    preferences, e.g. Bundle A contains Courses X, X, (list all courses
    in Bundle A). Course X is high preference, Course X is mid
    preference, Course X is low preference]
Bundle B: [First recall the courses in Bundle, then list matching
    preferences, e.g. Bundle B contains Courses X, X, (list all courses
    in Bundle B). Course X is high preference, Course X is mid
    preference, Course X is low preference]
</PREFERENCES>

<COMPLEMENTS>
Bundle A: [First recall the courses in Bundle, then list complementary
    relationships with magnitudes or "None", e.g. Bundle A contains
    Courses X, X, (list all courses in Bundle A). Course X and Course X
    are complements which helps moderately when taken together, Course
    X, Course X, and Course X are complements and helps significantly
    when taken together]
Bundle B: [First recall the courses in Bundle, then list complementary
    relationships with magnitudes or "None", e.g. Bundle B contains
    Courses X, X, (list all courses in Bundle B). Course X and Course X
    are complements which helps moderately when taken together, Course
    X, Course X, and Course X are complements and helps significantly
    when taken together]
</COMPLEMENTS>

<SUBSTITUTES>
Bundle A: [First recall the courses in Bundle, then list substitute
    relationships with magnitudes or "None", e.g. Bundle A contains
    Courses X, X, (list all courses in Bundle A). Course X and Course X
    are substitutes which harms moderately when taken together, Course
    X, Course X, and Course X are substitutes and harms significantly
    when taken together]
Bundle B: [First recall the courses in Bundle, then list substitute
    relationships with magnitudes or "None", e.g. Bundle B contains
    Courses X, X, (list all courses in Bundle B). Course X and Course X
    are substitutes which harms moderately when taken together, Course
    X, Course X, and Course X are substitutes and harms significantly
    when taken together]
</SUBSTITUTES>

<REASONING>
[Provide your concise reasoning in a few sentences, e.g. From the above,
    in terms of preferences, Bundle X is better. In terms of the
    presence and magnitude of complements, Bundle X is better. In terms
    of magnitude and precense of substitutes, bundle X is better.
    Considering the tradeoffs, Bundle X is better.]
</REASONING>

<CHOICE>Bundle X</CHOICE>
'''
```

## C.4 EXAMPLE OF A SYNTHETIC STUDENT

Note: the actual synthetic student in the experiment is stored as a python class object with the below values in the data structure. The natural language text below is just a text that converts the object into readable format.

```
Course Preferences (ordered from highest to lowest value within each
    tier):

High Priority Courses:
    1. Course 20 (value: 117.96)
    2. Course 19 (value: 115.30)

Medium Priority Courses:

Lower Priority Courses:
    1. Course 21 (value: 56.26)
    2. Course 7 (value: 55.99)
    3. Course 14 (value: 55.73)
    4. Course 17 (value: 55.71)
    5. Course 25 (value: 54.02)
    6. Course 2 (value: 52.66)
    7. Course 22 (value: 52.37)
    8. Course 3 (value: 52.07)
    9. Course 6 (value: 51.39)
    10. Course 8 (value: 50.80)
    11. Course 16 (value: 50.80)
    12. Course 1 (value: 49.36)
    13. Course 9 (value: 46.22)
    14. Course 13 (value: 45.73)
    15. Course 23 (value: 45.31)
    16. Course 15 (value: 45.18)
    17. Course 18 (value: 43.96)
    18. Course 4 (value: 43.71)
    19. Course 24 (value: 42.64)
    20. Course 5 (value: 42.55)
    21. Course 10 (value: 42.34)
    22. Course 12 (value: 41.71)
    23. Course 11 (value: 40.76)

Course Relationships:
Overlapping Content (Substitutes) - These are courses that HARMS YOU
    when taken together compared to taking only one, and the more the
    worse:
- Courses 19, 13, 16 overlap in content. Taking any two reduces their
    combined value by 40%, and taking all 3 reduces their combined value
    by 72%
- Courses 20, 25, 3, 17, 11 overlap in content. Taking any two reduces
    their combined value by 40%, taking any 3 reduces their combined
    value by 72%, taking any 4 reduces their combined value by 98%, and
    taking all 5 reduces their combined value by 118%

Complementary Courses (Complements) - These are courses that BENEFITS
    YOU HARMS YOU when taken together compared to taking only one, and
    the more the better:
- Courses 19, 12 complement each other. Taking any two increases their
    combined value by 40%
- Courses 20, 12, 5, 9 complement each other. Taking any two increases
    their combined value by 40%, taking any 3 increases their combined
    value by 72%, and taking all 4 increases their combined value by 98%
```

### C.5 SYNTHETIC STUDENT RESPONSE AND EFFECT OF CONCISENESS ON PERFORMANCE

#### C.5.1 NATURAL LANGUAGE DESCRIPTIONS OF PREFERENCES

The natural language descriptions provided by the student are a critical component of our framework. Below is an example of a detailed, baseline response using the prompt in Section C.2:

**Relatively Comprehensive (∼285 words):** : *"As I plan my courses for the upcoming semester, my top priorities are Course 4 and Course 24. Course 4 really stands out to me because it aligns perfectly with my major and offers a deep dive into topics I'm passionate about, plus it has a complementary relationship with other courses that can enhance my learning experience. Course 24 is also high on my list due to its intriguing content and potential to open up new career paths. In the medium priority tier, I'm eyeing Course 11 and Course 19. Course 11 seems like a solid choice to build on my existing knowledge, while Course 19, although slightly lower in value, could offer new perspectives that complement my high-priority courses. I'm aware that Course 19 overlaps with some of my other interests, so I'll need to be strategic about how I pair these with my other selections. When considering course combinations, I'm particularly mindful of the overlapping content and complementary courses. For example, Courses 24, 22, 19, 20, and 8 overlap, which means taking more than one could significantly reduce their combined value. Taking two is already suboptimal, and adding more courses from this group would diminish their effectiveness even further, with the most extreme case being taking all five, which should really be avoided. On the other hand, Courses 24, 9, 14, 6, and 15 complement each other, offering increased value when taken together. Even pairing two of these courses can provide a noticeable boost, and the more I can fit into my schedule, the better. Similarly, Courses 4, 19, and 1 work well in combination, enhancing each other's value. I need to balance these synergies with the overlaps to maximize my learning outcomes."*

This comprehensive response effectively conveys the student's preferences and provides sufficient detail to inform our mechanism. However, in practice, students might not always provide responses at this level of detail. To understand how response conciseness impacts our framework's performance, we introduce two alternative levels of brevity:

- **Moderately Brief:** Responses are prompted to be limited to approximately 180 words. While the LLM may not always adhere exactly to this constraint, the intent is to capture a moderately detailed response.

- **Highly Brief:** Responses are prompted to be limited to approximately 80 words. Again, the LLM might not strictly follow this word limit, but the responses are qualitatively much shorter and less detailed.

These levels are tested by asking the LLM proxy simulating student responses to adhere to the respective word constraints. Below are examples of the moderately brief and highly brief responses:

**Moderately Brief (∼180 words):** *"I'm really interested in taking Course 21, as it has the highest value among all the options. If I can't take it, my next choice would be Course 13. I also want to take Course 8, which is my only medium-priority course. Within the lower-priority tier, I'm interested in Course 6, Course 16, and Course 2. I'm aware that Course 13 and Course 3 have overlapping content, so taking both would be suboptimal. Similarly, taking Course 21, Course 11, Course 2, and Course 18 together would also be suboptimal. On the other hand, taking Course 13, Course 8, Course 24, and Course 5 together would be beneficial, as it would increase the combined value. The same applies to taking Course 21, Course 12, Course 8, Course 5, and Course 22 together.*

*In terms of bundles, I'd like to avoid taking Course 13 and Course 3 together, as it would be suboptimal. Taking both would be bad, taking all three (Course 13, Course 3, and Course 2) would be even worse, and taking all four (Course 13, Course 3, Course 2, and Course 10) would be really bad. I'd also like to avoid taking Course 21, Course 11, Course 2, and Course 18 together, as it would be suboptimal. Taking any two would be bad, taking all three would be worse, and taking all four would be really bad. On the other hand, taking Course 13, Course 8, Course 24, and Course 5 together would be beneficial, as it would increase the combined value."*

**Highly Brief (∼80 words):** *"My top priority courses are Course 11 and Course 18. I'm interested in Course 19, Course 16, and Course 22, but they're not as crucial. I also consider Course 21, Course 13, and Course 20 as potential options. For Course 6, Course 25, and Course 5, I'm not as*

*enthusiastic. Course 23 and Course 2 are decent choices, but I'm not sure about Course 17, Course 15, and Course 1. Course 8 and Course 3 are somewhat appealing, but Course 9 and Course 24 are less desirable. Course 14, Course 12, Course 7, and Course 4 are not my first choices."*

These variations allow us to test the robustness of our framework when students provide less detailed descriptions.

### C.5.2  IMPACT OF BREVITY ON VALUE AND CORRECTNESS

Table 7 summarizes the performance for each level of brevity, including normalized allocated bundle values, correctness rates, and statistical significance metrics. The correctness rates show only minor differences between the baseline (72.3%), moderately brief (71.7%), and highly brief (68.04%) versions. However, these differences in correctness significantly impact the normalized allocated bundle value, dropping from 119% for the baseline to 110% for the moderately brief version and 106% for the highly brief version. Notably, while the moderately brief version still demonstrates statistically significant improvement, the highly brief version does not.

| Brevity Level | Normalized Value $\pm CI$ | CQ Correctness % | % of Runs Better | Worse | P-value |
|---|---|---|---|---|---|
| Baseline | 119.34 ±6.50 | 72.3 | 74 | 26 | $6.84 \times 10^{-8}$ |
| Moderately Brief | 110.35 ±8.52 | 71.7 | 58 | 42 | 0.021 |
| Highly Brief | 106.54 ±8.29 | 68.04 | 52 | 48 | 0.129 |

Table 7: Summary of performance metrics for varying levels of brevity.

We evaluate the framework's performance by measuring both the normalized allocated bundle value and the correctness of the LLM in answering comparison queries. Figure 5 plots the changes in normalized allocated bundle value as the number of elicited CQs increases, while Figure 6 shows the correctness of the LLM at answering comparison queries.

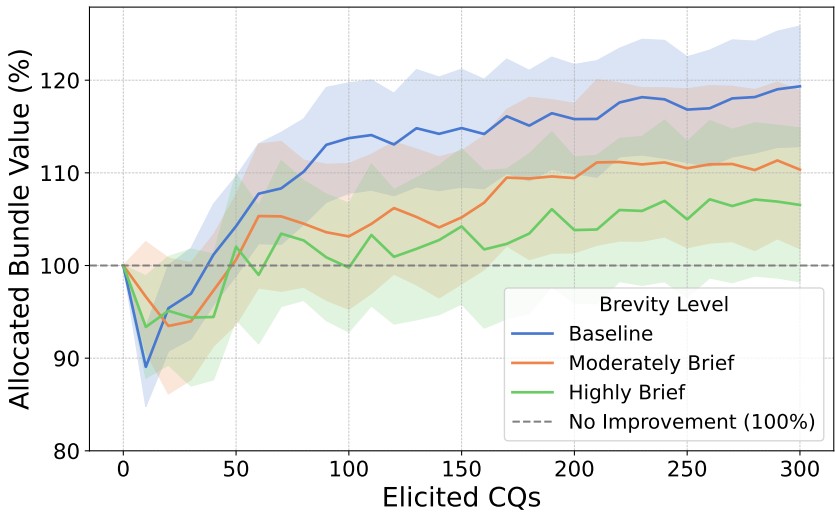

Figure 5: Normalized allocated bundle value as a function of elicited CQs for different brevity levels.

### C.5.3  DISCUSSION

These results suggest that our framework is robust to a degree when students provide less detailed responses. However, as the level of brevity increases, the framework's performance deteriorates. This is intuitive since the mechanism relies on the student's textual descriptions to answer comparison queries accurately. To mitigate this, incentivizing students to provide thorough responses or leveraging more advanced language models with better correctness guarantees can help maintain high performance.

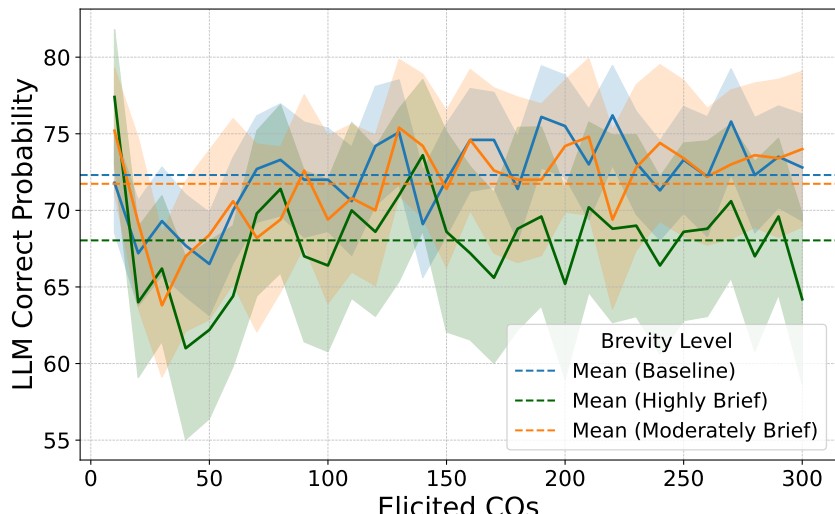

Figure 6: Correctness rate of the LLM at answering comparison queries for different brevity levels.

### C.6 DISCUSSION

These results suggest that our framework is robust to a degree when students provide less detailed responses. However, as the level of conciseness increases, the framework's performance deteriorates. This is intuitive since the mechanism relies on the student's textual descriptions to answer comparison queries accurately. To mitigate this, incentivizing students to provide thorough responses or leveraging more advanced language models with better correctness guarantees can help maintain high performance.

### C.7 EXAMPLES OF COMPARISON QUERY OUTPUT

The following is a correct answer for one CQ.

```
<PREFERENCES> Bundle A: Bundle A contains Courses 1, 9, 12, 22, 24.
    Course 22 is high preference, Course 9 is low preference, Courses 1,
    12, 24 are not mentioned in preferences. Bundle B: Bundle B contains
    Courses 8, 9, 12, 16, 22. Course 22 is high preference, Course 9 is
    low preference, Courses 8, 12, 16 are not mentioned in preferences.
    </PREFERENCES>

<COMPLEMENTS> Bundle A: Bundle A contains Courses 1, 9, 12, 22, 24.
    Course 22 and Course 12 are complements which helps significantly
    when taken together, Course 9 and Course 22 are complements which
    helps moderately when taken together. Bundle B: Bundle B contains
    Courses 8, 9, 12, 16, 22. Course 22 and Course 12 are complements
    which helps significantly when taken together, Course 9 and Course
    16 are complements which helps significantly when taken together.
    </COMPLEMENTS>

<SUBSTITUTES> Bundle A: Bundle A contains Courses 1, 9, 12, 22, 24.
    Course 22 and Course 12 are substitutes which harms moderately when
    taken together, Course 9 and Course 12 are substitutes which harms
    moderately when taken together. Bundle B: Bundle B contains Courses
    8, 9, 12, 16, 22. Course 22 and Course 12 are substitutes which
    harms moderately when taken together, Course 9 and Course 12 are
    substitutes which harms moderately when taken together.
    </SUBSTITUTES>

<REASONING> From the above, in terms of preferences, both bundles are
    similar since both contain high and low preference courses. In terms
```

```
      of the presence and magnitude of complements, Bundle B is better as
      it has two significant complements. In terms of magnitude and
      presence of substitutes, both bundles are similar. Considering the
      tradeoffs, Bundle B is better. </REASONING>

<CHOICE>Bundle B</CHOICE>
```

The following is an incorrect answer for the CQ. The difference between the two bundles is minimal, and the LLM simply compared the number of areas where each bundle excels.

```
<PREFERENCES> Bundle A: Bundle A contains Courses 9, 12, 14, 17, 21.
   Course 14 is high preference, Course 9 is low preference. Bundle B:
   Bundle B contains Courses 2, 14, 17, 21, 22. Course 14 and Course 22
   are high preference. </PREFERENCES>

<COMPLEMENTS> Bundle A: Bundle A contains Courses 9, 12, 14, 17, 21.
   Course 14 and Course 9 are complements which helps significantly
   when taken together. Course 12 and Course 14 are complements which
   helps significantly when taken together. Bundle B: Bundle B contains
   Courses 2, 14, 17, 21, 22. Course 14 and Course 22 are complements
   which helps significantly when taken together. </COMPLEMENTS>

<SUBSTITUTES> Bundle A: Bundle A contains Courses 9, 12, 14, 17, 21.
   Course 14 and Course 12 are substitutes which harms moderately when
   taken together. Bundle B: Bundle B contains Courses 2, 14, 17, 21,
   22. Course 14 and Course 22 are substitutes which harms
   significantly when taken together. </SUBSTITUTES>

<REASONING> From the above, in terms of preferences, Bundle B is
   better. In terms of the presence and magnitude of complements,
   Bundle A is better. In terms of magnitude and presence of
   substitutes, Bundle A is better. Considering the tradeoffs, Bundle A
   is better. </REASONING>

<CHOICE>Bundle A</CHOICE>
```

# D DETAILS FROM SECTION 3

## D.1 NEURAL NETWORK ARCHITECTURE AND TRAINING ALGORITHM DETAILS

Similar to Soumalias et al. (2024a), for the baseline architecture of the student's ML model, we used MVNNs (Weissteiner et al., 2022a). These are a special neural network architecture designed to model monotone value functions in combinatorial assignment problem. The training algorithm that we used to integerate both the student's GUI reports and the LLM-answered CQs is the algorithm proposed in Soumalias et al. (2024a), as that algorithm was designed to trained on exactly the same dataset types as we have.

## D.2 HYPERPARAMETER OPTIMIZATION (HPO)

In this section, we describe our hyperparameter optimization (HPO) methodology and the parameter ranges used.

Given the computational cost of querying LLMs, we employed *simulated LLMs* during the HPO process. First, we measured the accuracy of our default LLM architecture (as shown in Figure 4). During HPO, we simulated LLM responses by providing both correct and incorrect answers with probabilities that reflected the measured accuracy, assuming i.i.d. mistakes across queries.

The training algorithm described in Soumalias et al. (2024a) begins by training a student's ML model on her reports in the mechanism's original language, creating a cardinal dataset from those reports. The model is then fine-tuned based on student-answered CQs. In our framework, however, LLMs perform this task instead of students. Therefore, we optimized hyperparameters related to training on CQs and the acquisition function responsible for selecting which CQs the LLM answers.

Our optimization goal was to maximize the student's allocated bundle value, which serves as the primary performance metric. For each configuration tested, we ran separate instances and averaged performance over 10 runs, using different seeds from those employed in the experiments presented in Section 3.

The best-performing configuration and the full range of hyperparameters are provided in Table 8.

| Hyperparameter | HPO-Range | Winning Configuration |
|---|---|---|
| LLM CQ Batch Size | [1, 2, 4, 8, 16, 32] | 1 |
| LLM CQ Epochs | [2, 5, 10, 20, 50, 100, 200, 500, 1000] | 10 |
| LLM CQ Learning Rate | (0.0001, 0.1) | 0.01 |
| LLM CQ Weight Decay | [0, 1e-8, 1e-6, 1e-5, 1e-4, 1e-3, 1e-2, 1e-1] | 0.01 |
| LLM CQ Gradient Clipping | [0.01, 0.02, 0.05, 0.1, 0.2, 0.5, 1, 2, 5, 10] | 0.2 |
| LLM Generalized Cross Entropy $q$ | (0.001, 1.0) | 0.3 |
| LLM CQs | [200, 300, 500] | 500 |
| LLM Acquisition Function | [Boltzmann, Infomax, Random, Double TS] | Double TS |

Table 8: HPO ranges and winning parameters for our framework.

### D.3 DETAILED LEARNING EXPERIMENTS

In this section, we report a more detailed version of Table 2 from section Section 3.3.

We observe that our framework causes a dramatic improvement in learning performance for *all* metrics and quantiles tested.

| Metric | Whole Test Set | | Top 10% Quantile | | Top 5% Quantile | |
|---|---|---|---|---|---|---|
| | **Before** | **After** | **Before** | **After** | **Before** | **After** |
| $\text{MAE}_C$ | 46.14 | 30.69 | 51.23 | 27.96 | 51.15 | 26.19 |
| KT | 0.266 | 0.389 | 0.121 | 0.283 | 0.102 | 0.258 |
| $\text{MSE}_C$ | 3475 | 1942 | 4172 | 1500 | 4139.6 | 1309.4 |
| $R_C^2$ | −0.62 | 0.22 | −2.50 | 0.14 | −3.52 | 0.11 |

Table 9: Comparison of learning metrics before and after the LLM-elicited CQs. Each metric is reported for the whole test set, the top 10% quantile, and the top 5% quantile in terms of student value. Metrics include MAE, Kendall's Tau (KT), Mean Squared Error Centered (MSE-C), and R-squared centered ($R_C^2$). Shown are averages over 100 runs. Winners based on a paired t-test with $\alpha = 1\%$ are marked in grey.

## E ETHICS STATEMENT

This work aims to improve preference elicitation for combinatorial assignment problems, such as course allocation, by reducing the cognitive burden on users. This may lead to more equitable and efficient outcomes by allowing users to better express their preferences. Our experiments use a simulator based on prior lab data, and no new data was collected from human subjects. A potential risk is that an LLM proxy could misinterpret a user's preferences, leading to a poor allocation. Future real-world deployment would require rigorous user validation to mitigate this risk.

## F REPRODUCIBILITY STATEMENT

Our theoretical framework is detailed in Section 2, with proofs in Appendix B. The experimental setup, simulator parameters, and all LLM prompts are described in Section 4 and the appendices. The source code to reproduce all results is included in the supplementary material and will be made public, and we included it in the submission.

## G    LLM USAGE

LLMs were used as assistive tools for editing, grammar, and code debugging during the preparation of this manuscript. The core research ideas, theoretical framework, experimental design, and analysis were conceived and executed by the human authors, who reviewed and take full responsibility for all content.

