# OpenReview forum: "LLM-Powered Preference Elicitation in Combinatorial Assignment"
_ICLR.cc/2026/Conference — Submitted to ICLR 2026_

### Official Review · Reviewer_dzr7 · 2025-10-29

**Soundness:** 2
**Presentation:** 1
**Contribution:** 2
**Rating:** 4
**Confidence:** 3

**Summary:**

The paper proposes a one-shot preference elicitation method for combinatorial course allocation problem. Instead of asking students to answer many pairwise comparisons, each student provides a single free-text description of their preferences. A large language model then serves as a proxy, answering comparison queries during an active learning procedure. A neural value model is trained from these LLM-labeled comparisons using a noise-robust loss, and the final allocation is computed from the learned value functions. Experiments in a course allocation simulator show improved welfare relative to existing systems while requiring far less user effort.

**Strengths:**

The idea of using LLM proxies to reduce cognitive burden in interactive preference elicitation is interesting and well motivated. It leverages both the widespread availability of LLMs and their strong ability to interpret natural language descriptions of preferences. The course allocation setting is an appropriate application for this study. It is a natural problem and inherently combinatorial, which makes a good case study for preference elicitation problems facing similar combinatorial challenges.

**Weaknesses:**

- There is no clear description of the course allocation problem. Does the mechanism assign courses to multiple students at the same time, subject to course capacity constraints and the students' preference constraints? Or does it just select a bundle of courses for one student based on their preference?

- While the goal of reducing cognitive burden of repeate queries to the user is well motivated, the paper goes to the other extreme that almost completely removes interaction between the user and the mechanism. I'm not sure that this would be a good solution in practice, as it is very likely that the user does not include every detail about their preference in their initial text description, and become aware of additional preference constraints only when provided something that they dislike.

- The paper is too sketchy about some of the technique details. Sections 2.3.2 and 2.3.3 only give plain descriptions of the methods without any explanation to help understand the rationale behind.

**Questions:**

- Can you explain how the course allocation problem is defined? What does the mechanism output?

- Can you provide some intuitions about the methods and results in Sections 2.3.2 and 2.3.3?

- Proposition 2.1 seems to imply that the algorithm is able to recover the user's true value function. It seems very counterintuitive as we would expect the user's text description only reveal very limited information about their preference over all possible bundles. How should this result be interpreted? Besides that, how many comparisons are needed for the covergence?

- Why is only pair-wise interactions considered in Table 1? I suppose that it is likely that there are constraints like "I don't want all my courses to be theoretical", where more than two courses are interdependent.

---

### Official Review · Reviewer_VCQ9 · 2025-10-30

**Soundness:** 3
**Presentation:** 2
**Contribution:** 2
**Rating:** 2
**Confidence:** 3

**Summary:**

This paper explores solutions that utilize LLMs to help simplify preference elicitation (PE) in combinatorial assignment. Specifically, it proposes a framework that utilizes LLMs as proxies to collaborate with Machine learning-powered preference elicitation schemes.
The LLM proxies answer queries for humans guided by a small amount of textual human input.

This framework is applied and studied in the well-studied course allocation problem. Experiments evaluate the efficiency of LLM proxies against human questions. The results demonstrate that the proposed framework can enhance allocative efficiency across various LLMs.

**Strengths:**

+ The major motivation of the method is to address the issues faced by (a) the traditional structured reporting language, one-shot approach, and (b) iterative approaches.  The proposed method utilizes one-shot natural language input, eliminating the need for users to employ structured reporting language. From natural language input, an LLM proxy can extract answers to answer comparison queries (CQs) that are implemented within the framework. This eliminates the need for users to manually answer such CQs.

+ The loss function takes into consideration the noisy approximations from LLM proxies.

+ The experiments used a well-established simulator to generate data.

+ The experimental results show several important aspects of the work including (1) the framework's effectiveness, (2) the reason that the Double Thompson Sampling acquisition function is chosen, (3) effect of the noise factor, (4) robustness of the method w.r.t. the use of different LLMs.

**Weaknesses:**

W1. I am not convinced that a one-shot input approach is appropriate. The one-shot input works when a user is very clear about what type of information is useful for the system to know. Assuming a user is new to the system and has no prior knowledge of what to enter, it would be beneficial to allow users to fine-tune their input through multiple iterations. In other cases, users may just not be sure what their preferences are, and they want to "add" more constraints after their initial input or remove some constraints. A reasonable system should allow users to make such changes.

W2. The technical novelty is limited.
    (1) The proposed framework builds largely upon existing research, particularly the MLCM mechanism proposed by Soumalias et al. (2024a). This work extends the MLCM mechanism to allow one-shot natural language input. All the other components are nearly identical.
    (2) The proxy training utilizes both noise-robust loss and acquisition functions, which have been proposed previously. What does the novel idea come from in the training? Is it a novel idea to combine the loss and acquisition functions? Or, is it very challenging to combine the noise-robust loss and the acquisition function? If yes, please elaborate more to help readers understand the novelty.

W3. The manuscript's readability needs to be improved.
  The ML-powered learning component is never explained. Although the component was proposed in previous work, it should be briefly explained here for self-containment purposes.
  Terminologies in the specific course-matching context (e.g., queries, ML, value functions, acquisition functions) need to be explained first before their extensive use.
  The title says that the mechanism is combinatorial alignment. However, the work is actually only for the course allocation problem. It is best to change the title to the specific problem.

W4. The experiments convert each synthetic student's cardinal preferences into one-shot text input. However, in real life, it may not be possible to generate such one-shot text input as explained in W1. I suspect that the increase in performance is attributed to the fact that the one-shot has all the information.

W5. No real data is used to test the proposed method.

W6. A lot of important information about the experimental design is missing.

    - It is not clear how many synthetic students' data were generated and basic statistics of their cardinal preferences (e.g., average # of preferences for each student, the types of preferences).

    - For each synthetic student, how many 5-course combinations are presented? The experimental setting mentioned infinity. In the experiments, how many of them are actually calculated?

**Questions:**

Q1. How is the allocation value calculated?

Q2. Section 2.3.1: How is a student's ML model trained?

Q3. Figure 2a: How is the normalized allocated bundle value calculated? How do you get 19.3%?

Q4. Section 3.2: How do you know that the MLCM increases allocative efficiency by 14.2% using 20 student-answered CQs?

---

### Official Review · Reviewer_98cb · 2025-11-01

**Soundness:** 2
**Presentation:** 2
**Contribution:** 2
**Rating:** 4
**Confidence:** 4

**Summary:**

This paper proposes using Large Language Models (LLMs) as proxies for human preference elicitation in combinatorial assignment. This paper specifically focuses on course allocation as its main use case. The key problem it tries to address is that instead of requiring students to answer multiple comparison queries iteratively, let LLMs do the job. So first, students provide a single natural language description of their preferences, which an LLM proxy uses to answer queries on their behalf. The framework combines this with ML-powered preference learning using Monotone Value Neural Networks (MVNNs), Double Thompson Sampling for query selection, and Generalized Cross-Entropy loss for noise-robust training. Experiments on a course allocation simulator show that this approach achieves up to 20% improvement in allocative efficiency compared to baseline GUI-only methods, outperforming the state-of-the-art MLCM mechanism that requires 20 human-answered queries. The framework proves robust across different LLM architectures (ChatGPT, LLaMA) with 72-75% proxy accuracy, and maintains performance even with varying levels of student reporting errors. Theoretical guarantees show that when LLM accuracy exceeds 50%, the true value function minimizes expected GCE loss. The one-shot natural language approach reduces cognitive burden on users while improving allocation outcomes. Results demonstrate improvements across both allocation efficiency metrics and ML model learning performance, with particularly strong gains in high-value bundle regions.

**Strengths:**

1. The paper presents a standard solution, the user will present one structured query, and LLMs will do the iterative elicitation based on the query. The approach is natural and user-friendly, but quite well studied in the literature. The use of Double Thompson Sampling for efficient query selection that doubles performance versus alternatives, and epistemic MVNNs for uncertainty quantification, is nice.
2. The experimental evaluation is satisfactory. Testing robustness across multiple dimensions: different LLM architectures, varying accuracy levels (60-75%), different student error profiles, and various acquisition functions. The use of a validated simulator fitted to real-world data (Budish & Kessler 2021) adds to the experimental rigor, and results show statistical significance with proper confidence intervals.

**Weaknesses:**

1. The novelty is not clear. There are several papers in the literature that use TS (or variants of TS) for query selection. Some ablation study are missing.
2. Theoretical Analysis is Shallow: Proposition 2.1 only provides a weak guarantee—that the true value function is a minimizer of expected GCE loss when accuracy >50%. This doesn't bound approximation error, convergence rate, or sample complexity. There's no analysis of how many queries are needed, how accuracy requirements scale with problem complexity, or finite-sample guarantees. The connection to active learning theory and PAC-learning bounds is unexplored despite being highly relevant.
3. All experiments rely on synthetic students and simulated preferences. While the simulator is calibrated to real data, the paper lacks any human subject studies to validate that (a) real students would provide natural language descriptions of comparable quality, (b) LLM interpretations align with actual human preferences, and (c) students would be satisfied with allocations based on LLM proxies. The gap between simulation and deployment is substantial for a mechanism intended for real-world use.

**Questions:**

1. In your Proposition 2.1, it shows that the true value function minimizes expected GCE loss when LLM accuracy exceeds 50%, but provides no convergence rate or sample complexity analysis. Can you provide finite-sample bounds showing how many LLM-answered queries are required to achieve ε-optimal allocations as a function of bundle space size, LLM accuracy, and preference complexity? How does this compare theoretically to the query complexity of MLCM with human responses?
2. Natural language descriptions are inherently ambiguous and may not fully specify preferences over the exponential bundle space. How do you ensure consistency when the LLM proxy must extrapolate to bundles not clearly specified in the text? For example, if a student mentions "I prefer courses close together" but doesn't specify all pairwise preferences, how does the proxy handle novel bundle comparisons? Can you quantify what fraction of the bundle space is "covered" by typical natural language descriptions versus requiring LLM inference/hallucination?
3. Your entire evaluation uses LLMs to both generate natural language descriptions from synthetic preferences as well as answer queries based on those descriptions—essentially testing if an LLM can reconstruct its own outputs. I think this might not reflect real human queries. A human ablation study is at least a minimum requirement.
4. Why DTS? Have you tried other acquisition functions? Why not acquisition functions like BADGE, BAI,T which are more prevalent in deep active learning?

---

### Official Review · Reviewer_uMSy · 2025-11-02

**Soundness:** 3
**Presentation:** 3
**Contribution:** 3
**Rating:** 2
**Confidence:** 4

**Summary:**

This paper explores using large language models (LLMs) as one-shot proxies for human preference elicitation (PE) in combinatorial assignment problems, where traditional methods are cognitively demanding and limited in expressiveness. The proposed framework translates a natural language description of user preferences into structured inputs for allocation mechanisms, using GCE loss and Double Thompson Sampling for robustness and active learning. Experiments on course allocation (with real-world-fitted simulators) show up to 20% improvement in allocative efficiency over strong baselines, consistent across model types and input quality levels.

**Strengths:**

This paper introduces LLMs as low-effort, natural-language-based proxies for combinatorial PE.

The idea is supported by solid empirical results, with up to 20% efficiency improvement and robustness across models.

**Weaknesses:**

This paper is evaluated on a course-allocation simulator, a well-established benchmark for non-monetary combinatorial assignment problems. The focus on fairness and welfare is consistent with this domain, and the absence of pricing or strategic bidding mechanisms is therefore within the paper’s intended scope.

However, the evaluation could still benefit from broader validation. For example, by applying the framework to other assignment contexts, such as housing allocation or resource scheduling, to demonstrate robustness across varied combinatorial settings.

Moreover, the authors defer human-subject validation to future work, leaving the current evaluation dependent on simulated users and LLM proxies. This raises concerns about simulation bias and limits external validity. Even a small-scale user study, as in Christiano et al. (NeurIPS 2017) or Huang et al. (ICLR 2025), would provide valuable calibration of proxy fidelity. Overall, the contribution is promising but experimentally narrow, lacking evidence of generalization across wider problem settings or real human behavior.

Paul F. Christiano, Jan Leike, Tom B. Brown, Miljan Martic, Shane Legg, and Dario Amodei. “Deep Reinforcement Learning from Human Preferences.” NeurIPS 2017.

David Huang, Francisco Marmolejo-Cossío, Edwin Lock, and David C. Parkes. “Accelerated Preference Elicitation with LLM-Based Proxies.” ICLR 2025.

**Questions:**

An orthodox investigation of the proposed method for LLM-Powered Preference Elicitation is presented. What are the limits and caveats in the whole process in a world without human validation?

---

### Meta-Review · Area_Chair_9k2g · 2026-01-07

**Summary:**

The paper tackles an interesting problem of LLM-based preference elicitation in combinatorial assignment problems, but as the reviewers note there are several major weaknesses:
1. No human participant evaluation
2. Limited scope of application (one setting, course allocation)
3. Limited depth of the theoretical analysis
4. Limited novelty
5. Additional, more minor issues

**Reviewer Concerns:**

Unfortunately, there was no author response; all of the reviewers’ concerns stand.

**Reviewer Scores:**

Again, since there was no rebuttal, there was never an opportunity to participate in the discussion and thus no impetus for changes in scores.

---

### Decision · Program_Chairs · 2026-01-26

Reject